# Identifying key determinants and dynamics of SARS-CoV-2/ACE2 tight interaction

**Van A. Ngo**[1]\*, **Ramesh K. Jha**[2]\*

**1** Center for Nonlinear Studies, Los Alamos National Laboratory, Los Alamos, New Mexico, United States of America, **2** Bioscience Division, Los Alamos National Laboratory, Los Alamos, New Mexico, United States of America

\* rjha@lanl.gov (RKJ); ngov@lanl.gov (VAN)

## Abstract

SARS-CoV-2 virus, the causative agent of Covid-19, has fired up a global pandemic. The virus interacts with the human receptor angiotensin-converting enzyme 2 (ACE2) for an invasion via receptor binding domain (RBD) on its spike protein. To provide a deeper understanding of this interaction, we performed microsecond simulations of the RBD-ACE2 complex for SARS-CoV-2 and compared it with the closely related SARS-CoV discovered in 2003. We show residues in the RBD of SARS-CoV-2 that were mutated from SARS-CoV, collectively help make the RBD anchor much stronger to the N-terminal part of ACE2 than the corresponding residues on RBD of SARS-CoV. This would result in a reduced dissociation rate of SARS-CoV-2 from human receptor protein compared to SARS-CoV. The phenomenon was consistently observed in simulations beyond 500 ns and was reproducible across different force fields. Altogether, our study adds more insight into the critical dynamics of the key residues at the virus spike and human receptor binding interface and potentially aids the development of diagnostics and therapeutics to combat the pandemic efficiently.

## Introduction

It is vital to understand the mechanisms of SARS-CoV-2 in comparison to other coronaviruses [1–8]. One strategy is to zoom in to the molecular levels of how SARS-CoV-2 invades human cells. The first step of the invasion is to have its spike proteins bind to human cells such as lung cells, whose surface expresses a lot of ACE2. The RBD of SARS-CoV-2 [2] (dubbed RBD2 in this study) was identified to bind to ACE2 with an interface similar to the RBD of SARS-CoV (dubbed RBD1 hereafter) from 2003 [9]. Monoclonal antibodies from Covid-19 patients were found to directly interfere with the binding interaction between RBD2 and ACE2 [10]. It was determined that RBD2 binds stronger to ACE2 than RBD1 [4, 11, 12]. Multiple mutations are found in RBD2 that can explain why SARS-CoV-2 is more infectious than SARS-CoV. Specifically, residues G482/V483/E484/G485/F486/Q493/L455/N501 (based on RBD2 numbering) constituting a receptor binding motif (RBM) may cause a significant structural difference when comparing the crystal structures of RBD2 and RBD1 [2]. Another thorough set of Ala-

**Funding:** The research was supported by the DOE Office of Science through the National Virtual Biotechnology Laboratory, a consortium of DOE

national laboratories focused on response to COVID-19, with funding provided by the Coronavirus CARES Act. V.A.N is a Director's Postdoctoral Fellow at LANL and is partially funded by Laboratory Directed R&D Postdoctoral Research and Development fellowship (20170692PRD4). The funders had no role in study design, data collection and analysis, decision to publish, or preparation of the manuscript.

**Competing interests:** The authors have declared that no competing interests exist.

**Abbreviations:** ACE2, angiotensin-converting enzyme 2; covid, coronavirus disease; MD, molecular dynamics; PMF, potential of mean force; RBD, receptor binding domain; RBM, receptor binding motif; SARS, severe acute respiratory syndrome; FF, forcefield; US, umbrella sampling.

scanning simulations [13], pinpointed many critical residues, that can reduce the binding of RBD2 to ACE2. Free-energy calculations were also done for the mutations at these residues to compare how they contribute to overall binding affinities via a coarse grain model [14]. Using homology models for comparing the structures of Bat-CoV, SARS-CoV, and SARS-CoV-2, Ortega et. al. proposed that the two loops of RBD2 might help to enhance its interactions with ACE2 with an improvement of 1.6 kcal/mol in the binding energy compared to RBD1 [5]. Amin et al. showed this increase in the binding affinity may be attributed to the electrostatic interactions enhanced by the mutations appearing in RBD2 [15]. Spinello et al. identified various hydrogen bonds that reinforce the interactions in the RBD2-ACE2 complex compared to the RBD1-ACE2 complex [16]. This resulted in about 21 kcal/mol more binding free energy estimated via Molecular Mechanics Generalized Born Surface Area method [17]. To aid the research community, the Shaw group [7] performed 10–75 μs molecular dynamics (MD) simulations of both RBD1 and RBD2 with ACE2 and has granted free access to many critical simulation data. However, how these residues change the conformations at the RBD2-ACE2 interface in comparison to the RBD1-ACE2 interface remains to be elucidated. Even though free-energy calculations [14] reveal relatively important levels of these residues at the RBD2-ACE2 interface, how they dynamically co-operate to enhance the binding of RBD2 and how RBD2 may dynamically dissociate from ACE2 have not been described earlier.

In this study, we demonstrated how the key residues F486/N487/Y489/A475, some of which were previously identified as important, play a more critical and collective role at the RBD2-ACE2 interface than other residues (for example, Q493/L455/N501) and the corresponding residues L486/N487/Y489/P475 at RBD1-ACE2 interface (**Fig 1A**). We cross-examined these residues in two popular force fields (FFs), CHARMM36 [18] and AMBER FF [19]. Understanding how these groups of residues dynamically and collectively respond to an applied "force" at the RBD2-ACE2 interface may add more insight into the strategy of effective therapeutic design to specifically target these residues for battling the invasion of SARS-CoV-2 to the human host cells. To show such a collective response, we performed Umbrella Sampling (US) simulations of 4.8 μs in total per complex to compute the potential of mean forces (PMFs) for separating the RBDs from ACE2. Combining the results from long equilibration simulations and US simulations, we demonstrated that the dynamics of the residues at RBD1-ACE2 and RBD2-ACE2 interfaces are quantitatively different, that is, the loop, where F486/N487/Y489/A475 are located in RBD2, anchors strongly on ACE2, but the corresponding loop in RBD1 may lose its grip on ACE2.

## Methods

For the MD simulations, we used the crystal structure (PDB: 2AJF) [9] for the RBD1-ACE2 complex and the crystal structure (PDB: 6M0J) [11] for the RBD2-ACE2 complex (**Fig 1B**). Note that the residue numbers in **Fig 1A** are consistent with the alignment in Ref. [2, 4, 6]. For simplicity and consistency, we refer to the same residue numbers for both complexes. The residue indices in the PDB: 2AJF can be converted to the residue numbers shown in **Fig 1A** by adding 13 to indices less than 472 or by adding 14 to the indices otherwise since RBD2 showed an insertion of a residue at position 483. The missing loop (residues 375–381) in RBD1 was added by aligning RBD1 with RBD2; then the coordinates of the aligned loop (residues 389–394) were used to construct the coordinates for the missing loop in RBD1. We then used CHARMM 36 (C36) [18] force field (FF) to model the complexes. When building the models, we did not include glycans in either of the complexes. In the RBD1-ACE2 structure, the asparagine (N90) linked glycan (on ACE2), close to the interface, showed an extended form consisting of three sugar units (PDB: 2AJF). In RBD2-ACE2, N90 linked glycan showed a large

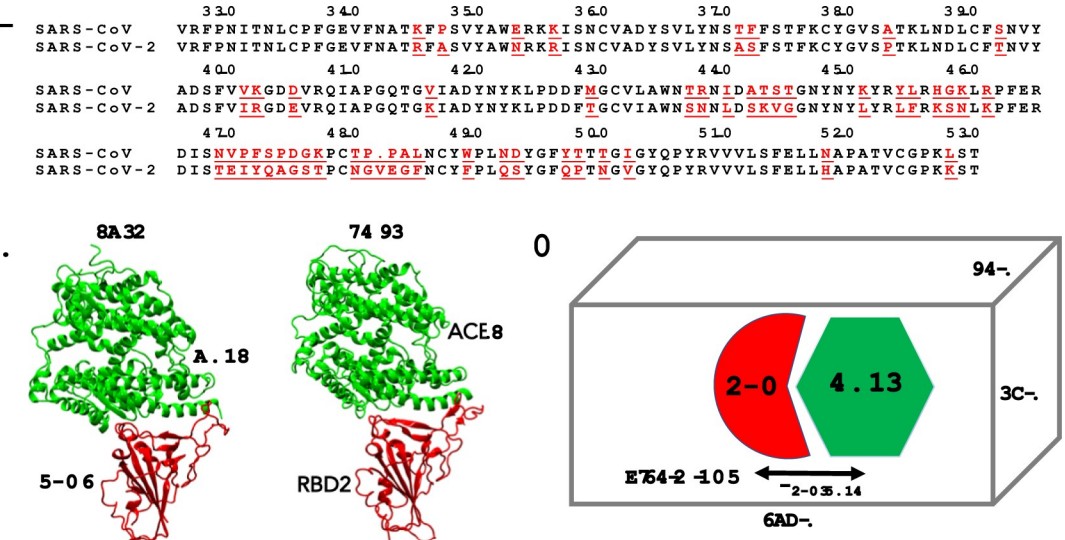

**Fig 1.** (A) Sequence alignment of receptor binding domains (RBD) of SARS-CoV and SARS-CoV-2. The residues underlined are mutations found in the RBDs. Any residue numbers referred in the text are positions in this sequence alignment. (B) X-ray structures of RBD1 of SARS-CoV (PDB: 2AJF) and RBD2 of SARS-CoV-2 (PDB: 6M0J), respectively, bound to human receptor angiotensin-converting enzyme 2 (ACE2). (C) MD simulation setup for RBD1-ACE2 and RBD2-ACE2 complexes.

discrepancy from structure to structure. While we saw only one sugar unit covalently attached to the N90 in two cases (PDBs: 6M0J and 7KMB), another structure of the RBD2-ACE2 complex (PDB: 6M17) showed two sugar units. Finally, the third form of glycosylation was also observed in a structure with three sugar units linked to N90 (PDB: 6VW1). Except for the latter one, in all RBD2-ACE2 complexes, the nearest amino acid to the glycan sugar T415 was >8 Å and expected to insignificantly contribute to the binding interaction. The structures of RBD-ACE2 complexes (PDB: 2AJF for RBD1-ACE2 & 6VW1 for RBD2-ACE2) that show long glycan on N90 also showed similar interaction with proximal residues T415 and R408 on RBD. Considering the two complexes were expressed using the same insect cell lines and 6VW1 experimental structure was determined by molecular replacement using 2AJF as structural template, we could not negate the effect of expression system on glycan pattern [20] and structural biases of solved structure towards the molecular replacement template. Further, mutation of N90 to eliminate glycosylation was shown to improve spike protein-mediated human cell infection [21], which is contrary to the observation based on interdomain contacts from the structure. Glycan linked to N322 on ACE2 also showed >8 Å distance from the nearest amino acid on the RBD. We also compared these dynamics modeled by the CHARMM force field with the model of the complexes using the AMBER force field in the presence of N-glycans. In the dynamics of the complexes using the AMBER force field, the N-glycan side-chains were at least 9 Å away from the residues of the RBD at the interfaces (**S1 Fig**). Due to the overall discrepancy in glycan from one structure to another and a low binding energy contribution expected from shorter glycan, the complexes were prepared in the absence of them. We included the $Zn^{2+}$ ion, which is coordinated by two histidine residues (H356 and H360) and a glutamate (E384) of ACE2. All histidine residues were modeled in the neutral and non-protonated state. We applied a patch of disulfide bond for all disulfide pairs as found in both RBDs and ACE2 [2, 9]. The systems were then solvated with the TIP3P water model [22]. 0.15 M KCl was added to neutralize and mimic a physiological condition. The number of $K^+$ and $Cl^-$ were kept at 102 and 78, respectively, for simulations of both RBD1-ACE2 and

RBD2-ACE2 complexes. The total number of atoms in these two solvated systems was approximately 102,000 with a dimension of $75 \times 83 \times 162$ Å$^3$ after minimization and 0.5 ns equilibration using NAMD 2.12 [23]. The minimization step was done with the conjugate gradient energy minimization method [24]. A cut-off distance of 12 Å and a switch distance of 10 Å was used to compute Lennard-Jones interactions. We enabled the Particle Mesh Ewald summation method [25] to compute electrostatic interactions with a grid size of 1.0 Å. We used Langevin dynamics together with Langevin Piston to keep the temperature around 310 K and pressure at 1 atm. A timestep = 1 fs was used for the short equilibration simulations.

To enable a larger integration timestep, we used PARMED [26] to re-partition hydrogen atoms bonded to protein heavy atoms (exclude water molecules), while keeping the total mass unchanged [27]. In this way, the dynamics of an entire protein may not noticeably change but the simulations can become more stable when using a timestep of 4 fs. We used AMBER version 16 [28, 29] to run efficiently on the GPU nodes using the same C36 FF and simulation parameters used in NAMD 2.12. Specifically, we can pack four replicas per GPUs-node without a cost of slowing down each of the four simulations per node via AMBER, while the same setup for the simulations using NAMD-GPU (timestep = 2 fs) can experience four times slower than AMBER-GPU (timestep = 4fs) on a test system of ~200k atoms, using NVIDIA Tesla P100 SXM2 GPGPUs. Since it has been shown that running multiple independent MD simulations in parallel can sample better than running a single long MD simulation [30, 31], we created 8 replicas of each complex, which were quickly equilibrated using NAMD 2.12. Each replica was further equilibrated using AMBER 16 for about 600 ns to collect data, which were saved every 1 ns for analysis. We collected data for a total of approximately 5 μs per complex.

To compute a potential of mean force (PMF), which can be used to describe an effective interaction between the complexes as a function of the distance between the two centers of mass, we performed Umbrella Sampling (US) simulations (**Fig 1B**) [32–34]. Typically, US simulations are prepared in a set of independent simulation windows, each of which has an applied or biased harmonic potential, e.g., $U_{US}(R_i,R) = k(R_i - R_{RBD-ACE2})^2/2$, where $k = 10$ kcal/mol-Å$^2$, $R_{RBD-ACE2}$ is the biased distance between the centers of mass of an RBD and of ACE2 (for simplicity, we used only Cα atoms to compute the centers of mass); and $R_i$ is a restraint position. We found the smallest value of $R_i$ from the distribution of $R_{RBD-ACE2}$ from the above 4.3 μs equilibration simulation. For instance, in case of the RBD1-ACE2 complex, the smallest value of $R_i$ is 43 Å; in case of the RBD2-ACE2 complex, the smallest value of $R_i$ is 46 Å. Then, we chose the last window with $R_i \sim 70$–75 Å to enable a complete separation between each RBD and ACE2. Since the harmonic potential induces a fluctuation of about 0.3–0.4 Å around each restraint position $R_i$, we used an increment of 0.5 Å between every adjacent window (**S2 Fig**) to ensure proper overlaps between the distributions of $R_{RBD-ACE2}$ in any two adjacent windows [35]. Each window was run for 10–11 ns per replica, i.e., for 8 replicas, we collected approximately 80 ns per window. In total, we simulated for 60 windows with a total of 4.8 μs for each complex. We saved the values of $R_{RBD-ACE2}$ every 1000 steps (= 4 ps) and the biased configurations every 1 ns for each window. These values were then used to solve for a potential of mean force via Weighted Histogram Analysis Method [33, 34]. To estimate uncertainty, we used different sets of trajectories as different blocks of data to compute standard deviations among the average values obtained from those sets.

## Results and discussions

### Key residues at RBD2-ACE2 interface

To rank the relative importance of each residue at the interfaces of RBD1 and RBD2 with ACE2, we computed the probability of each pair at the interfaces during the 5-μs equilibrium

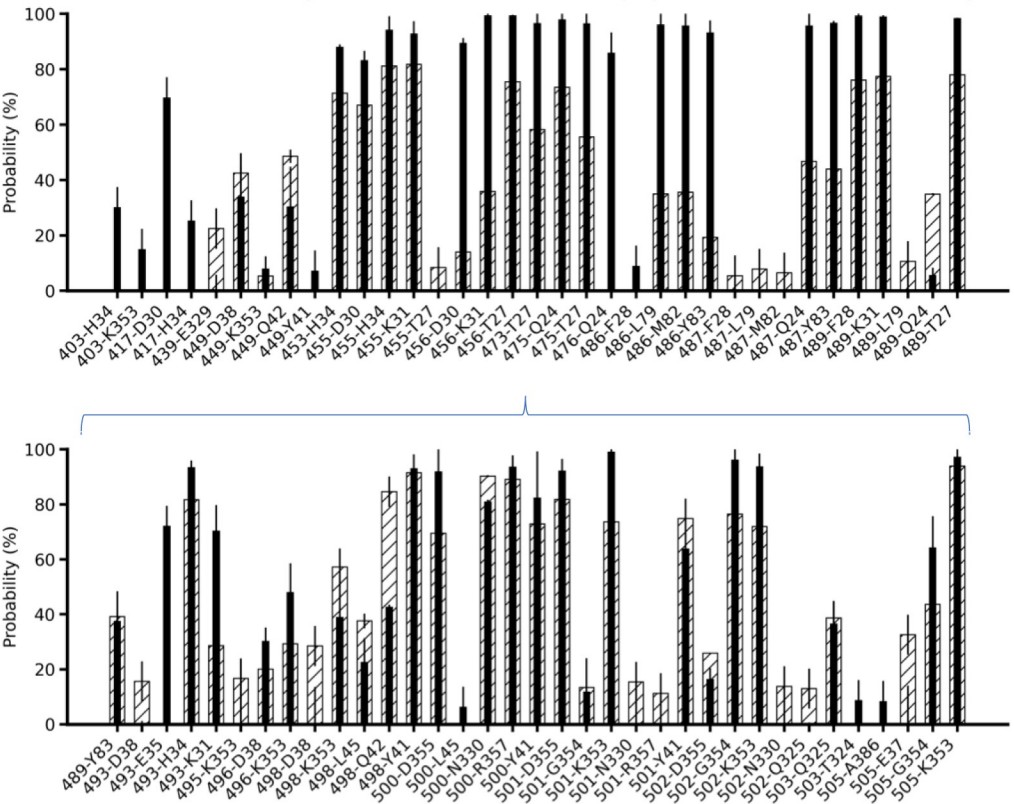

**Fig 2. Probability of aligned pairs of residues found within 3.5 Å at the RBD1-ACE2 (shaded bars) and RBD2-ACE2 (black bars) interfaces using C36 FF.** The first number corresponds to RBD based on alignment in Fig 1A. A stable pair is defined to have a probability larger than 60% during the simulation times. The clusters of improved interactions in RBD2-ACE2 are marked with braces.

simulations. **Fig 2** shows a relative ranking of pairs of residues at the interface: high probabilities indicate highly interactive pairs, while low probabilities indicate weakly interacting pairs. To the best of our knowledge, such a complete ranking has not been reported in previous studies, even though calculations of relative changes of interaction energies or free-energies were done for some of the pairs and mutations [14, 15]. **Fig 2** shows that both RBD1-ACE2 and RBD2-ACE2 interfaces have almost the same number of interacting pairs. However, if counting stable pairs that have probability larger than 60% (i.e., in close contact 60% of the total simulation time), there are only 22 ± 2 pairs in the RBD1-ACE2 interface, while RBD2-ACE2 showed substantially higher, 35 ± 1 stable pairs at the interface. Among these stable pairs, the mutations from RBD1 to RBD2, namely, Y455L, L456F, Y498Q, T501N and L486F appear to noticeably enhance the probabilities of interaction with the residues of ACE2 with multiple pairs having probabilities close to 100%. The brackets in **Fig 2** indicate two groups of neighboring residue clusters that have a substantial increase in interactions in the RBD2-ACE2 interface compared to the similar clusters in the RBD1-ACE2 interface. This suggests that they are key residues that differentiate the RBD2-ACE2 binding interface from the RBD1-ACE2 binding interface.

To probe whether the use of a different force field may change the outcomes of the key residues in the interfaces, we compared the probabilities with those obtained from the simulations using modified AMBER ff99SB FF [19] performed by the Shaw group [7]. In this AMBER-FF

model, the RBD1-ACE2 interface shows 29 ± 2 stable pairs, which are again substantially lesser than 39 ± 1 stable pairs in RBD2-ACE2 having close contact probabilities larger than 60% (**S3 Fig**). As a result, the AMBER FF quantitatively yields more stable pairs than the C36 FF. Particularly, the AMBER FF predicts two salt-bridges (R403-E37 and K417-D30) in the RBD2-ACE2 complex, compared to only one salt-bridge when modeled by the C36 FF (K417-D30). However, the Y455L, L456F, Y498Q, T501N and L486F mutations (**S3 Fig**) are also found to enhance the same clusters of interactions in the RBD2-ACE2 interface (**Fig 2**). Hence, regardless of the FFs, the RBD2-ACE2 interface shows stronger inter-molecular interactions than the RBD1-ACE2 interface involving the same group of residue positions (**Fig 2**). The clusters of the important residues are consistent with the finding [2] that residues L455 and F486 and N501 of SARS-CoV-2 are perhaps the most critical residues that increase the interactions between RBD2 and ACE2 in comparison with the interactions between RBD1 and ACE2 with Y455, L486 and T501 residues on RBD1. Note that L486F, Y498Q and D501 (instead of T501) to N mutations were found when the sequences of bat coronavirus RaTG13 and SARS-CoV-2 were compared [2]. Other key residues of RBD2 identified previously [2], are Q493 and S494 of which Q493 is also seen to have stable interactions with ACE2 (**Fig 2** and **S3 Fig**), while S494 was not detectable in both RBD1-ACE2 and RBD2-ACE2 interactions, suggesting that S494 may not play any critical role at the interface as proposed in the previous study. New residues such as K417, Y473 and A475, which were not reported earlier [2] but proposed to be important in others' work [5, 8, 15], emerged to be important for the RBD2-ACE2 interface and enhanced the interaction compared to RBD1-ACE2 with V417 and P475 residues. Particularly, the V417K mutation created a salt bridge with D30 (on ACE2), thus contributing to an increase in the electrostatic interactions [8] and compensating for coulomb energy due to mutation R439N and loss of distal R439-E329 electrostatic interaction (seen in RBD1-ACE2 interface) [15]. Overall, regardless of the FFs, there are the same key clusters of residues in the RBD2-ACE2 complex responsible for enhancing the interactions at the interface of the RBD2-ACE2 complex; however, the AMBER-FF model yielded a measurable stronger interface with a higher number of stable pairs than the C36-FF model. Such a difference might arise from different partial charges on protein atoms computed for AMBER-FF and C36-FF, a thorough comparison of which can be found elsewhere [36].

## Dynamics of RBD2-ACE2 and RBD1-ACE2 interfaces

To energetically differentiate the interfaces, we computed the probability $\rho(R_{\text{RBD-ACE2}})$ for a distance $R_{\text{RBD-ACE2}}$ between the center of mass of an RBD and ACE2; and a free energy was computed as $-k_{\text{B}}T\log(\rho(R_{\text{RBD-ACE2}}))$, where $k_{\text{B}}$ is the Boltzmann constant and $T = 310$ K is the simulated temperature. **Fig 3** shows the free-energy profiles computed for the RBD1-ACE2 and RBD2-ACE2 complexes using the two FFs. When modeled with the C36 FF, the global free-energy profile of RBD1-ACE2 complex with a global minimum (at $R_{\text{RBD-ACE2}} = 47$–48.5Å) was broader than the RBD2-ACE2 complex that showed a global minimum (at $R_{\text{RBD-ACE2}} \approx 50$ Å) (**Fig 3**). This suggests that the RBD1-ACE2 interface is less tight than the RBD2-ACE2 interface. When modeled by the AMBER FF, the global energy minima of the two complexes are also located at different positions ($R_{\text{RBD-ACE2}} = 44$ Å vs. $R_{\text{RBD-ACE2}} = 47$ Å for RBD1 and RBD2 respectively); and the shape of the free-energy profile of the RBD1-ACE2 complex for $R_{\text{RBD-ACE2}} > 44$ Å is approximately linear with $R$, in contrast to the parabolic shape of RBD2-ACE2 complex. In consistency with the C36-FF model, this suggests that the RBD2-ACE2 interface has stronger interactions than the RBD1-ACE2 interface. By comparing the free-energy profiles of each complex modeled by the two FFs, we observed that the complex modeled by the AMBER FF has stronger interactions than the complex modeled by the

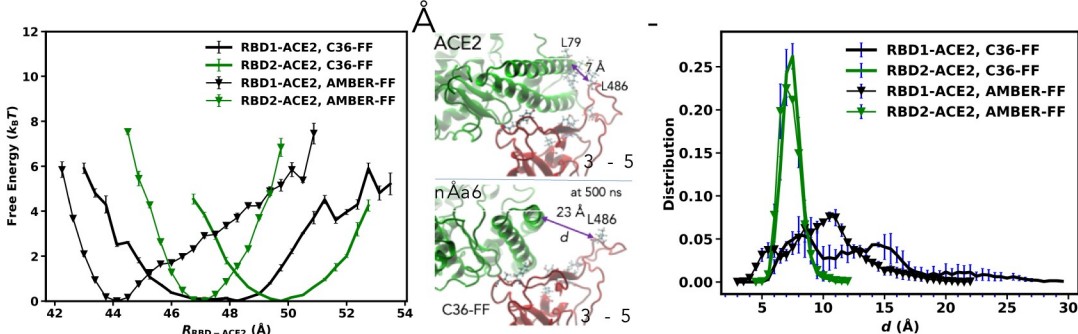

**Fig 3.** (A) Free-energy profiles computed as a function of the distance between the centers of mass of the RBDs and ACE2 using AMBER and C36 FFs. (B) Snapshots showing an initial configuration of RBD1-ACE complex and its configuration at 500 ns from the simulations using C36 FF. This 500 ns configuration was reproducible and showed in 3 out of 8 replicates (C) Distribution of the distance $d$ between the Cαs of L486 of RBD1 and L79 of ACE2 and compared with $d$ of F486-L79 in RBD2-ACE2.

C36 FF. For instance, the RBD2-ACE2 complex, if moved apart from the global minimum by 2 Å, shows a free energy increase of 4 $k_BT$ in the AMBER-FF model, but only about 2 $k_BT$ in the C36-FF model (**Fig 3A**). This is consistent with the above observation based on the pairs of residues at the interfaces (**Fig 2** and **S3 Fig**). To this end, even though we observed distinguishable features of the interactions modeled by the two FFs, one similarity stands out clear: the RBD2-ACE2 interface has stronger interaction energies than the RBD1-ACE2 interface as concluded in previous studies [2–5, 8, 11, 12, 14, 15, 37], which results in more stable pairs of interacting residues in the RBD2-ACE2 interface than the RBD1-ACE2 interface as observed in **Fig 2**.

Particularly for the RBD1-ACE2 complex, both FFs yield a meta-stable state, which is not evident in the RBD2-ACE2 complex. To the best of our knowledge, this observation has not been reported in any previous studies. To partially explain such meta-stable state observed in the C36-FF, **Fig 3B** shows a snapshot comparing the dynamical structure of RBD1 with its X-ray structure: initially, the loop having L486 and N487 is in close contact to the residues (L79, M82, Y83) of ACE2; at 500 ns, it loses the grip on ACE2 and moves up to 23 Å away from ACE2 residues (**S4 Fig**). This is not observed in the case of the RBD2-ACE2 complex, which remains similar to the crystal structure as shown in (**Fig 1B** and **S4 Fig**). To further quantify the behavior of this loop, we measured the distance $d$ between the Cα atoms of L486 in RBD1 (or F486 in RBD2) and L79 of ACE2 during the course of simulations. **Fig 3C** shows that in the case of the RBD1-ACE2 complex, while there is the highest peak in the distribution obtained from the AMBER-FF at $d$ = 11 Å, the broader distribution obtained from the C36-FF has two peaks at $d$ = 8 Å and $d$ = 15 Å, and non-zero values at $d$ ~25 Å. This is also consistent with the above observations that the C36-FF models weaker interactions, particularly between the RBD1 and ACE2 at $d$ ~25 Å than the AMBER-FF. In the RBD2-ACE2 complex, the distance is very stable at 7–7.5 Å as indicated by the very sharp peak in simulations using both FFs (**Fig 3C**). However, in the RBD1-ACE2 complex, the distributions shift to larger distances in case of both FFs. This suggests that the localized F486 at the RBD2-ACE2 interface may have a significantly stronger grip on ACE2 than the corresponding non-polar residue L486 in RBD1. This is also consistent with the profiles of solvent-accessible surface area (SASA) (**S5 Fig**) that the RBD1-ACE2 interface enable more accessible surfaces for solvent. By computing the Pearson correlation coefficient between $d$ and $R_{RBD-ACE2}$ we found a very small correlation (~0.2) between the movement of L486 with respect to the centers of mass. This might be due to the fact that the mass and motion of the loop having L486 might have little contribution to

the motion of the centers of mass. To this end, these results imply that the mutation L486F in RBD2 of SARS-CoV-2 might be the most consequential as F486 on SARS-CoV-2 has a more stable interaction with the ACE2 helices than L486 on SARS-CoV (**Fig 3B**); the L486 in RBD1 can move away from ACE2, resulting into less effective binding.

## Potential of Mean Force (PMF)

To shed more insight into the dissociation events of the RBDs from ACE2, we computed the PMFs of the two complexes via umbrella sampling (US) simulations using the C36 FF (see Methods). Note that the free-energy profiles shown in **Fig 3A** can be poorly sampled at distances away from the global minima, while the US sampling helps to improve rare statistics at those distances [32–34, 38, 39]. Since the comparison between the equilibrium simulations of RBD1/ACE2 and RBD2/ACE2 (**Fig 3**) yields similar differences for both force fields, it is reasonable to run umbrella sampling simulations for one force field, namely C36FF, to sample distances away from the global minima. **Fig 4A** shows that the PMFs for completely dissociating the RBDs from ACE2 around $R_{RBD-ACE2}$ = 70 Å are markedly different (the finite size effects [40] were not excluded from the profiles; so, there is a slight increase at the end of the curves due to the periodic images). We also observed that the meta-stable states in the RBD1-ACE2 complex (**Fig 3A**) become much shallower at $R$ = 52 Å (**Fig 4A**). The dissociation free energy, $\Delta G$ can be measured as the free-energy difference between $R$ = 70 Å and $R \sim 50$ Å, which is the global free-energy minimum. Our calculation of $\Delta G$ shows that it costs $\Delta G_1$ = 4.3 ± 0.8 and $\Delta G_2$ = 15.0 ± 0.8 kcal/mol for RBD1 and RBD2 respectively to completely dissociate from ACE2. In other words, the binding of RBD2 to ACE2 yields about 11 kcal/mol more energy than the binding of RBD1. This $\Delta\Delta G_{MD}$ (= $\Delta G1 - \Delta G_2$) is similar to an estimate (–13 kcal/mol) using metadynamics [41], but almost three times the value estimated by a coarse-grain model (–4.3 kcal/mol) in a recently published work [14], and half of the value estimated by Molecular Mechanics Generalized Born Surface Area method [17]. This disagreement may largely come from the simulated parameters (or FFs). It should be noted that $\Delta\Delta G_{MD}$ was computed without a thorough sampling of relative rotational, diffusion and conformational dynamics of the RBDs and ACE2, which might require much longer timescales than in this study. The calculations also assumed that the entire conformations of both RBDs and ACE2 are unchanged before the binding. These dynamics, if accounted for, may change the absolute value of $\Delta\Delta G_{MD}$, but likely $\Delta\Delta G_{MD}$ remains negative, indicating that the RBD2-ACE2 complex is tighter than the RBD1-ACE2 complex. Since the dissociation constant $K_d$ is proportional to

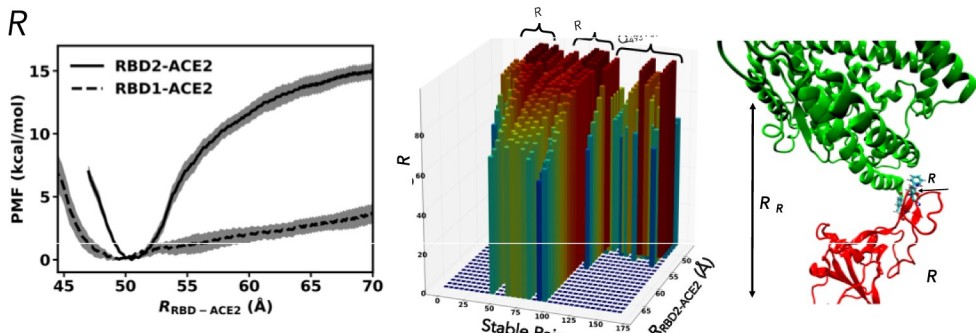

**Fig 4.** (A) Free-energy profiles computed from US simulations as function of $R_{RBD-ACE2}$ using C36-FF. (B) Three-dimension distributions of the stable pairs (refer Fig 2) as function of the biasing distance between the centers of mass of RBD2 and ACE2. (C) A snapshot during the Umbrella simulations using $R_{RBD-ACE2}$ = 70 Å. Residue A475 in RBD2 is located right behind F486.

exp($-\Delta G/R_{\text{B}}T$), where $\Delta G$ is the Gibbs free-energy difference between a binding state and unbinding state, $R_{\text{B}}$ is the gas constant, and $T$ is temperature, one can convert the experimental values of $K_{\text{d}}$ to the binding free energy difference $\Delta\Delta G_{\text{exp}}$ between RBD2-ACE2 and RBD1-ACE2 complexes to be about –0.9 to –1.5 kcal/mol [1, 2, 4, 11, 12, 42]. This is an order of magnitude smaller than $\Delta\Delta G_{\text{MD}}$. As we noted that $\Delta\Delta G_{\text{MD}}$ is not conceptually the same as $\Delta\Delta G_{\text{exp}}$. In fact, the association and dissociation rates measured in experiments are macroscopic effective rates [43–45], which describe a two-step process: a rotational and translational diffusion step drives particles into contacts, at which they can bind and unbind via an interaction potential with the intrinsic microscopic association and dissociation rates, respectively. An estimation [46] of $K_{\text{d}}$ from the PMF without addressing this two-step process is likely incorrect, even though it may quantitatively yield results similar to experimental measurements. We will address this in a separate study.

To determine which residues of RBD2 are critical and behave collectively during the relatively quick dissociation of the complex during the same timescales of few microseconds, we computed the probabilities of the stable residue pairs (**Fig 2**) at the interface as a function of the biasing distance between the two centers of mass of RBD2 and ACE2 (**Fig 4B**). **Fig 4B** shows that while the two clusters of residues (L455/F456/Y473 and Q493/T500/N501/G502/Y505) of RBD2 dissociate from ACE2 at $R$ = 55 Å (5 Å away from the global minimum), the cluster of residues F486/N487/Y489/A475 (**Fig 4C**) remains in contact with ACE2 with a probability of 60% at $R$ = 70 Å. Note that the similar pattern of the changes in the probabilities of these residues indicate that, this cluster of residues behaves as a "sticky" anchor of the spike protein of SARS-CoV-2 that holds on the binding region of ACE2. In contrast, as observed in **Fig 3B** and **3C**, the similar cluster of L486/N487/Y489/P475 in RBD1 does not have a strong grip on ACE2. These distinguishable outcomes suggest that the mutations L486F and P475A may play a critical role that makes RBD2 noticeably stickier to ACE2 than RBD1.

### Reversal mutations F486L and A475P on RBD2 to imitate RBD1 interface

To test whether single mutations F486L and A475P can reverse the ACE2-binding anchor of RBD2 to the RBD1 loop (**Fig 3C**), we ran similar 8-replicas simulations of the mutations (2×5 μs) and computed the distributions of the distance between the Cαs of the residue position 486 of the mutants and L79 of ACE2 in comparison with the distribution computed for the RBD2-WT. **Fig 5** shows that the mutations allow the mutated anchors to fluctuate slightly further away from the point of contact on ACE2. Specifically, F486L mutation allows the anchor to shift to a distance of 9 Å at a probability of 8 ± 1%, while the WT anchor has a probability of only 3± 1% at $d$ = 9 Å, even though the average values of $d$ are almost identical, 7.9 ± 0.1 Å. Compared to the distribution of RBD1-ACE2 in **Fig 3C**, the mutations do not cause a substantial change in the dynamics of the binding anchor. In other words, single mutations F486L and A475P, while slightly reducing interactions with ACE2, are not sufficient to reverse the binding anchor of RBD2. This suggests that perhaps the double mutations and the other mutations neighboring to these residues (**Fig 1A**) provide some structural restraints on the dynamics of the binding anchor of RBD2 as observed in **Fig 3C**.

### Conclusions

In this study, we present simulation data of approximately 30 μs in total (4×5 μs equilibration and 2×4.8 μs US simulations), which show key stable pairs between the SARS-CoV or SARS-CoV-2 RBDs (**Fig 2**) and the human receptor ACE2. We showed three clusters of residues that have enhanced interactions in RBD2 and ACE2. Particularly, one cluster of residues, including F486/N487/Y489/A475, is perhaps the most critical cluster of amino acids that appear to

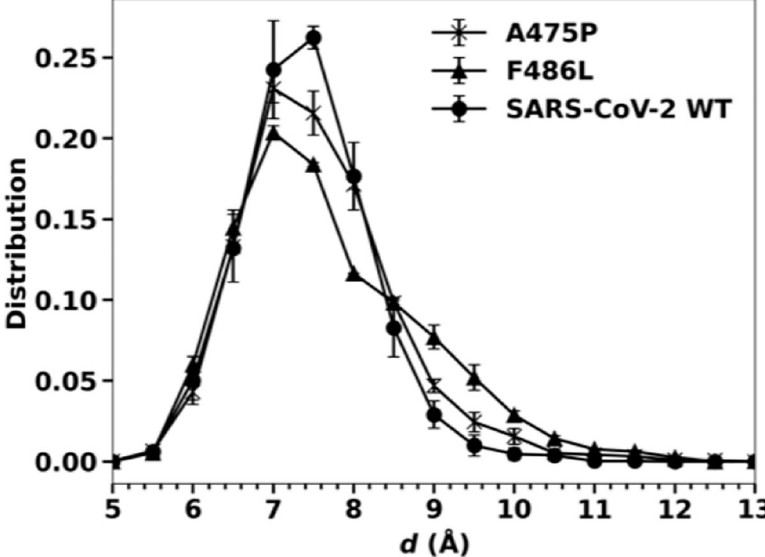

**Fig 5. Distribution of the distance $d$ between the C$\alpha$s of L486 of the F486L mutant and L79 of ACE2 and compared to $d$ between F486 of the A475P mutant and L79 of ACE2.**

collectively induce strong interactions between RBD2 and ACE2 (**Figs 3C** and **4**). This cluster of residues could act as a binding anchor that strongly holds on the N-terminal part of ACE2. In contrast, the corresponding set of residues L486/N487/Y489/P475 of RBD1 can loosen its grip from ACE2 (**Fig 3B and 3C**). The observations were consistent across different force-fields. The dramatic difference in the dynamics of these residue clusters is only observed in MD simulations of beyond 500 ns, which many previous studies [2–5, 8, 11, 12, 14, 15, 37] did not report. We also found that single mutations F486L and A475P do not reverse the dynamics of the RBD1 anchor. This suggests that the double mutations (L486F and P475A) and the other mutations neighboring to these residues (**Fig 1A**) might collectively provide some struc-tural restraints on the dynamics of the binding anchor of RBD2, so that this anchor can have a significantly stronger grip on ACE2.

Our work is consistent with another multi-microsecond study, which suggested that the rigidity of the RBD2-ACE2 complex is attributed to the π-stacking of F486 with Y83 that increases the stability of the loop (so-called loop $L_3$ having F486) compared to the RBD1-ACE2 complex [16], but did not point out a wild movement of the corresponding residue in the RBD1-ACE2 complex (**Fig 3**). Wang et al. [47] showed that the mutation F486L can reduce the overall binding affinity by 1.2 kcal/mol. This is also consistent with a recent study [13] examining Alanine mutations of the interfacial residues, which show that F486A may also dis-rupt the correlations of the residues at the RBD2-ACE2 interface and that N487A can increase the flexibility of the loop $L_3$.

During the completion of our work, newer variants of the SARS-CoV-2 evolved, that showed mutations in the spike protein including the RBD2. It is expected that some of them will perturb the interaction between ACE2 and spike protein either by creating new interac-tions, or by stabilizing the spike protein itself, as suggested from a recent published article [48]. Our finding showed N501 as an important residue of the cluster Q493/T500/N501/G502/Y505 that give binding advantage to RBD2 over RBD1 (**Fig 2**), but this cluster is also first to dissoci-ate from ACE2 (**Fig 4B**) indicating sub optimal interactions compared to other clusters. N501 has been identified in several new variants to be mutated to tyrosine. Mutations like N501Y is

expected to provide larger surface area for stabilizing the RBD2 or for interaction with ACE2. E484 and T478 were not identified as interfacial residues during our simulations, as they did not meet the cutoff distance (3.5 Å). But new mutations such as E484K and T478K on RBD2 in evolved SARS-CoV-2 variants are expected to bring these positions within the interface cut-off due to longer side chain of lysine and hence perturb the binding interaction. Understanding these mutations becomes valuable not only to identify the transmissibility of the SARS variant but also whether any therapeutic or diagnostic measure get compromised.

While this work was under completion, several work that probed the involvement of glycans for ACE2 recognition and binding by the SARS-CoV-2 spike protein got published. In a computational study, N322-linked glycan showed strengthening of the RBD2-ACE2 complex while N90 glycosylation could interfere with the RBD2-ACE2 interaction [49]. Another study has elucidated the contribution of different glycan types, and has suggested that distinct glycans can have subtle effects on RBD-ACE2 interaction [50]. Consistent with Ref. [50], an experimental investigation of glycosylated ACE2 showed very subtle effect (positive or negative depending on the glycoform) on binding to spike protein and removal of glycans from ACE2 showed very minimal perturbation to the binding [51]. Hence, our predictions solely looking at the residues at the protein interface in the absence of glycans, though physiologically less accurate, could still provide biophysically relevant insights to the viral attachment. It will be interesting though to look at the participating residues at the RBD2-ACE2 interface and dynamics of the interaction in the presence of glycans, especially since a detailed mapping of ACE2 glycosylation patterns is available now [52].

A contribution from our study is a comparison of the dynamics of the interfaces across two commonly used force fields, one of which, namely, C36 FF shows more dynamical behaviors of the interface than AMBER FF. Our estimated PMFs are markedly different from the reported PMFs computed for the same systems using a coarse-grain model [46]. In this coarse-grain model, the PMFs of both RBD1-ACE2 and RBD2-ACE2 as a function of the centers of mass between the proteins have an extremely narrow minimum at $R = 10$ Å and a free-energy barrier of about 11 kcal/mol at $R = 28$ Å. From a thermodynamic point of view, such a narrow free-energy minimum and high barrier may not indicate stable complexes as observed in our simulations. Collectively, these results build our understanding of key determinants of molecular recognition of SARS-CoV-2 and human receptor protein and provide opportunities to target the residue clusters for specific and sensitive binders [53] for therapeutics and diagnostics alike.

## Supporting information

**S1 Fig. Distance of N-glycans from the ACE2-RBD interface.** Snapshots from molecular dynamics simulations performed by the Shaw group (https://www.deshawresearch.com/downloads/download_trajectory_sarscov2.cgi/) showing distance of ACE2 N-glycans N322 and N53 from the nearest RBD1 (left, PDB code 2AJF) and the RBD2 (right, PDB code 6M17) residues. Other N-glycans on ACE2 and RBD are also shown and lie at a distance greater than 15 Å from the interface.
(PDF)

**S2 Fig. Distributions of $R_{RBD-ACE2}$ from the US sampling simulations.**
(PDF)

**S3 Fig. Probability of aligned pairs of residues found within 3.5 Å at the RBD1-ACE2 (shaded bars) and RBD2-ACE2 (black bars) interfaces using AMBER ff99SB FF performed by the Shaw group.** The first number corresponds to RBD based on alignment in Fig 1A. A

stable pair is defined to have a probability larger than 60% during the simulation times. NOTE: S19 interactions are split into two parts that include the terminal group of residue-18 on ACE2.
(PDF)

**S4 Fig. Time series of the distance d between the Cαs of L486 of RBD1 and L79 of ACE2 and F486 of RBD2 and L79 of ACE2.** Dashed lines indicate the starting point of independent simulations with the indicated numbers. The value of $d \geq 23$ (Fig 3B) occurs a few times in case of RBD1-ACE2. Overall, the interactions between L486 and L79 in RBD1-ACE2 complex is shown to be weaker than corresponding F486 and L79 in RBD2-ACE2.
(PDF)

**S5 Fig. Solvent accessible surface area (SASA) computed during an accumulated time of approximately 4 μs (0.5 μs/trajectory) for the initial residues at the interfaces of RBDs-ACE2.** VMD [DOI:10.1016/0263-7855(96)00018-5] was used to compute SASA for the interface residues. A script can be found here: https://www.ks.uiuc.edu/Research/vmd/mailing_list/vmd-l/att-18670/sasa.tcl.
(PDF)

**S1 File. Coordinates of RBD-ACE2 complexes for the snapshots during MD simulations illustrated in Fig 3B (0 & 500 ns) and Fig 4C.**
(ZIP)

**S2 File. Binary trajectory files of MD simulations for RBD1-ACE2 and RBD2-ACE2 complexes.**
(ZIP)

**S1 Movie. Movie (mpg) files of MD simulations for RBD1-ACE2 and RBD2-ACE2 complexes.**
(ZIP)

## Acknowledgments

The work was authored under Triad National Security, LLC ("Triad") Contract No. 89233218CNA000001 with the U.S. Department of Energy. This research used computational resources provided by the Los Alamos National Laboratory Institutional Computing Program (under w20_foldamers to R.K.J. and x20_simcovid19 to V.A.N.), which is supported by the U. S. Department of Energy National Nuclear Security Administration under Contract No. 89233218CNA000001. The authors will like to thank Drs. Kien Nguyen and Srirupa Chakraborty for fruitful discussions on the manuscript.

## Author Contributions

**Conceptualization:** Van A. Ngo, Ramesh K. Jha.

**Formal analysis:** Van A. Ngo, Ramesh K. Jha.

**Funding acquisition:** Ramesh K. Jha.

**Investigation:** Van A. Ngo.

**Resources:** Ramesh K. Jha.

**Software:** Van A. Ngo.

**Supervision:** Ramesh K. Jha.

**Writing – original draft:** Van A. Ngo.

**Writing – review & editing:** Ramesh K. Jha.

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
