## [Decision Letter · Decision Letter 0]

19 May 2021

PONE-D-21-09645

Identifying Key Determinants and Dynamics of SARS-CoV-2/ACE2 Tight Interaction

PLOS ONE

Dear Dr. Jha,

Thank you for submitting your manuscript to PLOS ONE. After careful consideration, we feel that it has merit but does not fully meet PLOS ONE’s publication criteria as it currently stands. Therefore, we invite you to submit a revised version of the manuscript that addresses the points raised during the review process.

The reviewers have several important points that I am asking you to consider in a revised version. I understand that performing additional calculations is not always practical, and if you choose to not perform some of the calculations that the reviewers are suggesting (MMPBSA / SASA etc) , please justify in your answer why it is not necessary. Please consider the request from the reviewers to discuss your results in the context of recent publications, in particular these that adress the role and importance of glycosylation of the protein.

We look forward to receiving your revised manuscript.

Kind regards,

Jerome Baudry, Ph.D.

Academic Editor

PLOS ONE

Additional Editor Comments:

The reviewers have several important points that I am asking you to consider in a revised version. I understand that performing additional calculations is not always practical, and if you choose to not perform some of the calculations that the reviewers are suggesting (MMPBSA / SASA etc) , please justify in your answer why it is not necessary. Please consider the request from the reviewers to discuss your results in the context of recent publications, in particular these that adress the role and importance of glycosylation of the protein.

Journal Requirements:

[Research was supported by the DOE Office of Science through the National Virtual Biotechnology Laboratory, a consortium of DOE national laboratories focused on responseto COVID-19, with funding provided by the Coronavirus CARES Act. V.A.N is a Director’s Postdoctoral Fellow at LANL and is partially funded by Laboratory Directed R&D Postdoctoral Research and Development fellowship (20170692PRD4). The work was authored under Triad National Security, LLC (“Triad”)Contract No. 38089233218CNA000001 with the U.S. Department of Energy. This research used computational resources provided by the Los Alamos National Laboratory Institutional Computing Program (under w20_foldamers to R.K.J. and x20_simcovid19 to V.A.N.), which is supported by the U.S. Department of Energy National Nuclear Security Administration under Contract No. 89233218CNA000001.]

 [Research was supported by the DOE Office of Science through the National Virtual Biotechnology Laboratory, a consortium of DOE national laboratories focused on response to COVID-19, with funding provided by the Coronavirus CARES Act.

The funders had no role in study design, data collection and analysis, decision to publish, or preparation of the manuscript.]

Additionally, because some of your funding information pertains to Triad National Security, LLC, we ask you to provide an updated Competing Interests statement, declaring all sources of commercial funding.

In your Competing Interests statement, please confirm that your commercial funding does not alter your adherence to PLOS ONE Editorial policies and criteria by including the following statement: "This does not alter our adherence to PLOS ONE policies on sharing data and materials.” as detailed online in our guide for authors  http://journals.plos.org/plosone/s/competing-interests.  If this statement is not true and your adherence to PLOS policies on sharing data and materials is altered, please explain how.

4. Thank you for submitting the above manuscript to PLOS ONE. During our internal evaluation of the manuscript, we found significant text overlap between your submission and the following previously published works, some of which you are an author.

https://www.cell.com/cell/fulltext/S0092-8674(20)30262-2?_returnURL=https%3A%2F%2Flinkinghub.elsevier.com%2Fretrieve%2Fpii%2FS0092867420302622%3Fshowall%3Dtrue

Please revise the manuscript to rephrase the duplicated text, cite your sources, and provide details as to how the current manuscript advances on previous work. Please note that further consideration is dependent on the submission of a manuscript that addresses these concerns about the overlap in text with published work.

Reviewers' comments:

Reviewer's Responses to Questions

**Comments to the Author**

1. Is the manuscript technically sound, and do the data support the conclusions?

Reviewer #1: Yes

Reviewer #2: Partly

2. Has the statistical analysis been performed appropriately and rigorously? 

Reviewer #1: Yes

Reviewer #2: Yes

3. Have the authors made all data underlying the findings in their manuscript fully available?

Reviewer #1: Yes

Reviewer #2: No

4. Is the manuscript presented in an intelligible fashion and written in standard English?

Reviewer #1: Yes

Reviewer #2: Yes

5. Review Comments to the Author

Reviewer #1: 1) Authors have not included any glycans in their simulations. Recent papers have shown that some of these glycans are important for binding (e.g., glycan – glycan interactions on the RBD – ACE2 binding). For example:

https://www.biorxiv.org/content/10.1101/2021.03.30.437783v1

https://www.biorxiv.org/content/10.1101/2020.07.09.193680v1.full

Authors should consider explaining their results in light of these papers and if possible, include how their results compare to studies conducted with glycans included.

Authors should briefly talk about potentially affect in the conformational dynamics of the protein when glycans are removed which can indirectly affect interactions?

2) Authors report that the results differ between the CHARMM and AMBER forcefields but they do not discuss possible reasons behind this observation. The authors should consider explaining the reasons behind the reported differences. Additionally, while most of the analyses in the manuscript were done using both the CHARMM and AMBER forcefields, the authors used only the CHARMM forcefield for PMF and the reasons behind the change were not stated. Few sentences explaining the rationale behind this would be useful.

3) Authors can consider performing MM-PBSA to get the binding energies and to also get the per residue energy decompositions.

4) Authors should consider calculating the number and lifetime of hydrogen bonds or hydrophobic interactions, etc., between ACE2-RBD1/2 as it would be more informative for interaction comparison.

5) Authors should define or explain what “bound” probability means in the result section. Was the probability just based on a distance cutoff? Or a hydrogen bond is being defined as interaction?

6) Author should report some statistic so as to show the convergence of their simulations.

7) Authors observed a minima shift of 2Å in Fig. 3, which seems very small relatively. Authors should explain the significance of these results in more detail.

8) Authors can also consider talking about water-mediated interactions at the interfaces.

9) Authors have reported a lot of interesting results, but they need to provide more explanation on how it connects to the biology and physiology of the disease. Especially with respect to the current variants, they may also suggest some mutants which they believe can make the binding stronger in the future, hence making the variant more lethal.

Reviewer #2: Comments for the Authors

The outbreak and spread of COVID-19 diseases caused by the severe acute respiratory syndrome coronavirus 2 (SARS-CoV-2) infection now is well-known as a global concern to the public health worldwide. Angiotensin-converting enzyme 2 (ACE2) is an enzyme attached to the cell membranes of cells located in multiple tissues including lungs, arteries, heart, kidney, and intestines. It also serves as the entry point into cells for some coronaviruses, including severe acute respiratory syndrome coronavirus (SARS-CoV), and SARS-CoV-2. In this study, in order to pinpoint key residues involved in the SARS-CoV receptor binding domain (RBD, RBD1)-ACE2 and SARS-CoV-2 receptor binding domain (RBD, RBD2)-ACE2 binding, Ngo et al. performed microsecond simulations for these two complexes using different force fields together with free energy calculation and mutation studies, indicating that F486/N487/Y489/A475 is perhaps the most critical cluster of amino acids that appear to collectively induce strong interactions between RBD2 and ACE2. Taken together, the microsecond direct simulations and Umbrella Sampling (US) simulations by defining a range of center-of-mass distances between RBD and ACE2 results provide insightful information pertaining to these indispensable residues responsible for their binding and would promote our understanding and facilitate future therapeutics and diagnostics strategical improvement, considering that multiple SARS-CoV-2 mutant strains were prevalent with stronger infections recently.

There are several points described below,

1.Did the authors provided the Supporting Information Figures? I didn’t find Supporting Information Figures of 1-3. It severely affects my understanding/evaluation of the major analyses and conclusions described in the main text, such as the overlaps of adjacent US windows.

2.Page 2, line 41, why was the “Spike protein” capitalized?

3.Page 3, line 68, Is “L486/N487/Y48/P475” a typo? According to the context and Fig1A, it should be “F486/N487/Y489/A475”.

4.There are several resolved complex structures of RBD1-ACE2 and RBD2-ACE2 as shown in Method section (page 4), do they have large differences? Why did the authors select 2AJF and 6M0J for the simulations?

5.The authors considered Zn2+ ions when building the whole simulation system, which seems located at the binding interfaces. However, the influence of Zn2+ ions were not studied and discussed.

6.When computing the potential of mean force (PMF), the authors only considered the distance between the two centers of mass. However, it can be seen from the Fig. 3B that ACE2 and RBD1 also showed rotational motion, which can be measured by defining an additional angle parameter. Taking both the distance and angle into the free energy calculation would provide more meaningful and comprehensive results.

7.I suggest that the authors also include the analysis of solvent-accessible surface area (SASA) monitor over time, which is closely associated with the molecular contacts of the interfacial residues.

8.Amber FF model yielded a measurable stronger interface with higher number of stable pairs than the C36-FF model. Why? Is it mainly attributed to their charge or vdW parameter difference?

9.I suggest the authors to use different colors for two force fields showing in the Fig. 3A/C as it is now quite difficult to tell which is which.

10.In page 10, lines 249-250. Where are results of the Pearson correlation coefficient calculations?

11.In page 11, lines 338-339, “The dramatic difference in the dynamics of these residue clusters is only observed in MD simulations of beyond 500 ns, which many previous studies [2–5,8,11,12,14,15,36] did not report.” There should be a Figure to support this observation.

12.In Fig. 4A, which force field was used for this analysis? In Fig. 4B, the color bar should be provided.

6. PLOS authors have the option to publish the peer review history of their article (what does this mean?). If published, this will include your full peer review and any attached files.

Reviewer #1: No

Reviewer #2: No

---

## [Author Response · Author response to Decision Letter 0]

8 Jul 2021

Response to Reviewers

Additional Editor Comments:

The reviewers have several important points that I am asking you to consider in a revised version. I understand that performing additional calculations is not always practical, and if you choose to not perform some of the calculations that the reviewers are suggesting (MMPBSA / SASA etc) , please justify in your answer why it is not necessary. Please consider the request from the reviewers to discuss your results in the context of recent publications, in particular these that address the role and importance of glycosylation of the protein.

 Reply: We put our best effort to address reviewer comments.

Journal Requirements:

Reply: We fixed the title page according to the guidelines.

[Research was supported by the DOE Office of Science through the National Virtual Biotechnology Laboratory, a consortium of DOE national laboratories focused on response to COVID-19, with funding provided by the Coronavirus CARES Act. V.A.N is a Director’s Postdoctoral Fellow at LANL and is partially funded by Laboratory Directed R&D Postdoctoral Research and Development fellowship (20170692PRD4). The work was authored under Triad National Security, LLC (“Triad”)Contract No. 38089233218CNA000001 with the U.S. Department of Energy. This research used computational resources provided by the Los Alamos National Laboratory Institutional Computing Program (under w20_foldamers to R.K.J. and x20_simcovid19 to V.A.N.), which is supported by the U.S. Department of Energy National Nuclear Security Administration under Contract No. 89233218CNA000001.]

 [Research was supported by the DOE Office of Science through the National Virtual Biotechnology Laboratory, a consortium of DOE national laboratories focused on response to COVID-19, with funding provided by the Coronavirus CARES Act.

The funders had no role in study design, data collection and analysis, decision to publish, or preparation of the manuscript.]

Reply: We moved the funding information from the Acknowledgement to Funding Statement

Additionally, because some of your funding information pertains to Triad National Security, LLC, we ask you to provide an updated Competing Interests statement, declaring all sources of commercial funding.

In your Competing Interests statement, please confirm that your commercial funding does not alter your adherence to PLOS ONE Editorial policies and criteria by including the following statement: "This does not alter our adherence to PLOS ONE policies on sharing data and materials.” as detailed online in our guide for authors http://journals.plos.org/plosone/s/competing-interests. If this statement is not true and your adherence to PLOS policies on sharing data and materials is altered, please explain how.

Reply: We provided the Competing Interests Statement as requested.

Reply: in the revised version, we provided a specific number for the benchmark statement, namely “the same setup for the simulations using NAMD-GPU (timestep = 2 fs) can experience four times slower than AMBER-GPU (timestep = 4fs) on a test system of ~200k atoms, using NVIDIA Tesla P100 SXM2 GPGPUs”

4. Thank you for submitting the above manuscript to PLOS ONE. During our internal evaluation of the manuscript, we found significant text overlap between your submission and the following previously published works, some of which you are an author.

https://www.cell.com/cell/fulltext/S0092-8674(20)30262-2?_returnURL=https%3A%2F%2Flinkinghub.elsevier.com%2Fretrieve%2Fpii%2FS0092867420302622%3Fshowall%3Dtrue

Please revise the manuscript to rephrase the duplicated text, cite your sources, and provide details as to how the current manuscript advances on previous work. Please note that further consideration is dependent on the submission of a manuscript that addresses these concerns about the overlap in text with published work.

Reply: The paper mentioned in the link was one of the first papers that talked about spike protein. We have cited this paper in our work. While we cannot negate the fact that our tone at some places could have been swayed by that paper, I assure you that there is no deliberate copying of text verbatim. Please help us (direct us) to the right paragraphs where you have been able to detect any overlapping extracts, if that is possible.

Reviewers' comments:

Reviewer's Responses to Questions

Comments to the Author

1. Is the manuscript technically sound, and do the data support the conclusions?

Reviewer #1: Yes

Reviewer #2: Partly

2. Has the statistical analysis been performed appropriately and rigorously? 

Reviewer #1: Yes

Reviewer #2: Yes

3. Have the authors made all data underlying the findings in their manuscript fully available?

Reviewer #1: Yes

Reviewer #2: No

4. Is the manuscript presented in an intelligible fashion and written in standard English?

Reviewer #1: Yes

Reviewer #2: Yes

5. Review Comments to the Author

Reviewer #1: 1) Authors have not included any glycans in their simulations. Recent papers have shown that some of these glycans are important for binding (e.g., glycan – glycan interactions on the RBD – ACE2 binding). For example:

https://www.biorxiv.org/content/10.1101/2021.03.30.437783v1

https://www.biorxiv.org/content/10.1101/2020.07.09.193680v1.full

Authors should consider explaining their results in light of these papers and if possible, include how their results compare to studies conducted with glycans included.

Authors should briefly talk about potentially affect in the conformational dynamics of the protein when glycans are removed which can indirectly affect interactions?

Reply: This is a very good point. When we initiated this work, there were only very limited number of structures of the RBD2-ACE2 complex available. Further, they showed discrepancy in their glycosylation pattern. We also found out in the literature that the protein expressed from a source, such as insect cells vs mammalian or human cells can perturb the glycosylation. Hence, in order to avoid biasing towards incorrect glycosylation and also since some of them were far from RBD2, we did not include them in our simulations. We have added this explanation in the Methods section.

“When building the models, we did not include glycans in either of the complexes. In the RBD1-ACE2 structure, the asparagine (N90) linked glycan (on ACE2), close to the interface, showed an extended form consisting of three sugar units (PDB: 2AJF). In RBD2-ACE2, N90 linked glycan showed a large discrepancy from structure to structure. While we saw only one sugar unit covalently attached to the N90 in two cases (PDBs: 6M0J and 7KMB), another structure of the RBD2-ACE2 complex (PDB: 6M17) showed two sugar units. Finally, the third form of glycosylation was also observed in a structure with three sugar units linked to N90 (PDB: 6VW1). Except for the latter one, in all RBD2-ACE2 complexes, the nearest amino acid to the glycan sugar T415 was >8 Å and expected to insignificantly contribute to the binding interaction. The structures of RBD-ACE2 complexes (PDB: 2AJF for RBD1-ACE2 & 6VW1 for RBD2-ACE2) that show long glycan on N90 also showed similar interaction with proximal residues T415 and R408 on RBD. Considering the two complexes were expressed using the same insect cell lines and 6VW1 experimental structure was determined by molecular replacement using 2AJF as structural template, we could not negate the effect of expression system on glycan pattern [20] and structural biases of solved structure towards the molecular replacement template. Further, mutation of N90 to eliminate glycosylation was shown to improve spike protein-mediated human cell infection [21], which is contrary to the observation based on interdomain contacts from the structure. Glycan linked to N322 on ACE2 also showed >8 Å distance from the nearest amino acid on the RBD. We also compared these dynamics modeled by the CHARMM force field with the model of the complexes using the AMBER force field in the presence of N-glycans. In the dynamics of the complexes using the AMBER force field, the N-glycan sidechains were at least 9 Å away from the residues of the RBD at the interfaces (S1 Fig). Due to the overall discrepancy in glycan from one structure to another and a low binding energy contribution expected from shorter glycan, the complexes were prepared in the absence of them.”

2) Authors report that the results differ between the CHARMM and AMBER forcefields but they do not discuss possible reasons behind this observation. The authors should consider explaining the reasons behind the reported differences. Additionally, while most of the analyses in the manuscript were done using both the CHARMM and AMBER forcefields, the authors used only the CHARMM forcefield for PMF and the reasons behind the change were not stated. Few sentences explaining the rationale behind this would be useful.

Reply: We added some discussions about the reasons of the difference between CHARMM and AMBER force field. One major reason is that both force fields use slightly different parametrization strategies (see DOI: 10.1002/jcc.20082). For instance, CHARMM and AMBER also use different approaches to compute charges for proteins: while CHARMM is based the QM calculations using HF/6-31G* for supramolecular data, AMBER is based on RESP (Restrained Electrostatic Potential) to fit to the QM calculations. A thorough comparison between AMBER and CHARMM can be found in the work by McKerell (DOI: 10.1002/jcc.20082). We used the D.E. Shaw’s simulation data with Amber forcefield (available in the link: https://www.deshawresearch.com/downloads/download_trajectory_sarscov2.cgi/). Our goal was to see if different force fields gave rise to different observation. We showed here that, in fact, different force fields yield differences in dynamics between RBD1/ACE2 and RBD2/ACE2 complexes (Fig 3). We only computed PMF for one force field for the complexes. We explained this in the main text, which reads “Since the comparison between the equilibrium simulations of RBD1/ACE2 and RBD2/ACE2 (Fig 3) yields similar differences for both force fields, it is reasonable to run umbrella sampling simulations for one force field, namely C36FF, to sample distances away from the global minima.”

3) Authors can consider performing MM-PBSA to get the binding energies and to also get the per residue energy decompositions.

Reply: MM-PBSA calculations and “the per residue energy decomposition” were computed in previous work, for instance, Zhang et al, (In silico binding profile characterization of SARS-CoV-2 spike protein and its mutants bound to human ACE2 receptor, Brief Bioinformatics, 2021 PMCID: PMC8194596). However, the published work did not report clear different dynamics happening at the interfaces. Hence, we focused our work around the dynamics of the protein-protein interaction.

4) Authors should consider calculating the number and lifetime of hydrogen bonds or hydrophobic interactions, etc., between ACE2-RBD1/2 as it would be more informative for interaction comparison.

Reply: This is a very good suggestion. However, these types of calculations have been reported elsewhere for shorter simulation timescales, for instance, see Ref. [2–5,8,11,12,14,15,36] in the text.. Since these calculations were computed mostly for shorter timescales, we only reported what have not been discussed: the difference dynamics at the interfaces during longer timescales. 

5) Authors should define or explain what “bound” probability means in the result section. Was the probability just based on a distance cutoff? Or a hydrogen bond is being defined as interaction?

Reply: We removed “bound” from the text. This is simply the probability computed based on distance cutoff. The implication of this probability is explained in the next sentence; it reads “high probabilities indicate highly interactive pairs, while low probabilities indicate weakly interacting pairs.” Yes, a hydrogen bond is included in this probability.

6) Author should report some statistic so as to show the convergence of their simulations.

Reply: Thank you for bringing this up. It is possible that the convergence has not reached. We aim to examine the dynamics of the interfaces in “longer” simulations compared to the presented simulations, which could shed more definitive answers for the convergence. Currently that is beyond the scope of this manuscript.

7) Authors observed a minima shift of 2Å in Fig. 3, which seems very small relatively. Authors should explain the significance of these results in more detail.

Reply: We thank the reviewer for pointing this out. There are a couple of types of the free-energy minimum shifts in Figure 3A. The first one is a shift of about 1.5 Å between RBD1-ACE2 and RBD2-ACE2 in the same force field. This one is related to different sequences of RBD1 versus RBD2 and the movement of the loops shown in Figure 3B. The second shift is due to difference in force field, that is, AMBER versus CHARMM. In AMBER force field the distances between RBDs and ACE2 are 2 Å closer than those in CHARMM force field. This indicates relatively stronger attractions between RBDs and ACE2 in AMBER force field. Also, the shapes of the profiles are visually broader in CHARMM than in AMBER. We discussed this point in the text, which reads “When modeled with the C36 FF, the global free-energy profile of RBD1-ACE2 complex with a global minimum (at RRBD-ACE2 = 47-48.5Å) was broader than the RBD2-ACE2 complex that showed a global minimum (at RRBD-ACE2 � 50 Å) (Fig 3). This suggests that the RBD1-ACE2 interface is less tight than the RBD2-ACE2 interface.” (Pg 9, Ln 242-246).

8) Authors can also consider talking about water-mediated interactions at the interfaces.

Reply: Water-mediated interactions are manifested in the pair-residue interactions and the free-energy profiles in Fig. 3A. We are also more interested in the conformations of the loops of RBDs as shown in Figure 3B.

9) Authors have reported a lot of interesting results, but they need to provide more explanation on how it connects to the biology and physiology of the disease. Especially with respect to the current variants, they may also suggest some mutants which they believe can make the binding stronger in the future, hence making the variant more lethal.

Reply: We added a paragraph to include some of the observed mutations in the RBD of the spike protein in the new variants of SARS-CoV-2. 

“During the completion of our work, newer variants of the SARS-CoV-2 evolved, that showed mutations in the spike protein including the RBD2. It is expected that some of them will perturb the interaction between ACE2 and spike protein either by creating new interactions, or by stabilizing the spike protein itself, as suggested from a recent published article [47]. Our finding showed N501 as an important residue of the cluster Q493/T500/N501/G502/Y505 that give binding advantage to RBD2 over RBD1 (Fig 2), but this cluster is also first to dissociate from ACE2 (Fig 4B) indicating sub optimal interactions compared to other clusters. N501 has been identified in several new variants to be mutated to tyrosine. Mutations like N501Y is expected to provide larger surface area for stabilizing the RBD2 or for interaction with ACE2. E484 and T478 were not identified as interfacial residues during our simulations, as they did not meet the cutoff distance. But new mutations such as E484K and T478K on RBD2 in evolved SARS-CoV-2 variants are expected to bring these positions within the interface cutoff due to longer side chain of lysine and hence perturb the binding interaction. Understanding these mutations becomes valuable not only to identify the transmissibility of the SARS variant but also whether any therapeutic or diagnostic measures get compromised.”

[47] Shorthouse D, Hall Benjamin A. SARS-CoV-2 Variants are Selecting for Spike Protein Mutations that Increase Protein Stability. Biophysics; 2021 Jun. doi:10.1101/2021.06.25.449882

Reviewer #2: Comments for the Authors

The outbreak and spread of COVID-19 diseases caused by the severe acute respiratory syndrome coronavirus 2 (SARS-CoV-2) infection now is well-known as a global concern to the public health worldwide. Angiotensin-converting enzyme 2 (ACE2) is an enzyme attached to the cell membranes of cells located in multiple tissues including lungs, arteries, heart, kidney, and intestines. It also serves as the entry point into cells for some coronaviruses, including severe acute respiratory syndrome coronavirus (SARS-CoV), and SARS-CoV-2. In this study, in order to pinpoint key residues involved in the SARS-CoV receptor binding domain (RBD, RBD1)-ACE2 and SARS-CoV-2 receptor binding domain (RBD, RBD2)-ACE2 binding, Ngo et al. performed microsecond simulations for these two complexes using different force fields together with free energy calculation and mutation studies, indicating that F486/N487/Y489/A475 is perhaps the most critical cluster of amino acids that appear to collectively induce strong interactions between RBD2 and ACE2. Taken together, the microsecond direct simulations and Umbrella Sampling (US) simulations by defining a range of center-of-mass distances between RBD and ACE2 results provide insightful information pertaining to these indispensable residues responsible for their binding and would promote our understanding and facilitate future therapeutics and diagnostics strategical improvement, considering that multiple SARS-CoV-2 mutant strains were prevalent with stronger infections recently.

There are several points described below,

1.Did the authors provided the Supporting Information Figures? I didn’t find Supporting Information Figures of 1-3. It severely affects my understanding/evaluation of the major analyses and conclusions described in the main text, such as the overlaps of adjacent US windows.

Reply: We’re sorry that the Supporting Information document was not obvious in our submission. We submitted the document earlier. We have a revised version now and named Plos_Supp_Info_final_1.3

2.Page 2, line 41, why was the “Spike protein” capitalized?

Reply: This was typographical error. We will put in lower case. 

3.Page 3, line 68, Is “L486/N487/Y48/P475” a typo? According to the context and Fig1A, it should be “F486/N487/Y489/A475”.

Reply: Thanks for pointing this out. We fixed this.

4.There are several resolved complex structures of RBD1-ACE2 and RBD2-ACE2 as shown in Method section (page 4), do they have large differences? Why did the authors select 2AJF and 6M0J for the simulations?

Reply: At the start of this project, these two structures were the highest resolution structure available, and hence we used them for our work.

5.The authors considered Zn2+ ions when building the whole simulation system, which seems located at the binding interfaces. However, the influence of Zn2+ ions were not studied and discussed.

Reply: No, Zn2+ is not located at the interface. It is located near the center of ACE2 far away from the interface and actually stabilizing the interdomain interface of ACE2. Others have shown that Zn2+ only fluctuates minimally during the course of MD simulations (DESRES-ANTON-10905033; https://www.deshawresearch.com/downloads/download_trajectory_sarscov2.cgi/).

6.When computing the potential of mean force (PMF), the authors only considered the distance between the two centers of mass. However, it can be seen from the Fig. 3B that ACE2 and RBD1 also showed rotational motion, which can be measured by defining an additional angle parameter. Taking both the distance and angle into the free energy calculation would provide more meaningful and comprehensive results.

Reply: We appreciate the suggestions, but we did not sample for relative angles between the two proteins. That would be way too expensive for this simple study of the interfaces of RBDs and ACE2.

7.I suggest that the authors also include the analysis of solvent-accessible surface area (SASA) monitor over time, which is closely associated with the molecular contacts of the interfacial residues.

Reply: We thank you for suggesting this. We computed SASA as a function of time, shown in S5 Fig in the supporting information. The time-series of SASA also confirm that the RBD1-ACE2 interface is much more flexible than the RBD2-ACE2 interface.

S5 Fig. Solvent accessible surface area (SASA) computed during an accumulated time of approximately 4 �s (0.5 �s/trajectory) for the initial residues at the interfaces of RBDs-ACE2. VMD [DOI:10.1016/0263-7855(96)00018-5] was used to compute SASA for the interface residues. A script can be found in this link: https://www.ks.uiuc.edu/Research/vmd/mailing_list/vmd-l/att-18670/sasa.tcl

8.Amber FF model yielded a measurable stronger interface with higher number of stable pairs than the C36-FF model. Why? Is it mainly attributed to their charge or vdW parameter difference?

Reply: The major differences between AMBER and C36FF include both charges and VDW parameters. At the interfaces, there are very few salt-bridges; so it is reasonable to assume that the VDW interactions in AMBER are stronger at the interface. We discussed this point in the text (Pg 9, Ln 231-232)

9.I suggest the authors to use different colors for two force fields showing in the Fig. 3A/C as it is now quite difficult to tell which is which.

Reply: We changed the color to make the presentation better. 

10.In page 10, lines 249-250. Where are results of the Pearson correlation coefficient calculations?

Reply: We provided the number (~0.2) for the Pearson correlation coefficient calculations.

11.In page 11, lines 338-339, “The dramatic difference in the dynamics of these residue clusters is only observed in MD simulations of beyond 500 ns, which many previous studies [2–5,8,11,12,14,15,36] did not report.” There should be a Figure to support this observation.

Reply: Figure 3B actually shows the snapshot at approximately 500 ns. In any event, we plot the time-series.

S4 Fig. Time series of the distance d between the C�s of L486 of RBD1 and L79 of ACE2 and F486 of RBD2 and L79 of ACE2. Dashed lines indicate the starting point of independent simulations with the indicated numbers. The value of d � 23 (Figure 3B) occurs a few times in case of RBD1-ACE2. Overall, the interactions between L486 and L79 in RBD1-ACE2 complex is shown to be weaker than corresponding F486 and L79 in RBD2-ACE2.

12.In Fig. 4A, which force field was used for this analysis? In Fig. 4B, the color bar should be provided.

Reply: We used CHARMM36. We replotted Fig. 4B with color bars.

6. PLOS authors have the option to publish the peer review history of their article (what does this mean?). If published, this will include your full peer review and any attached files.

Do you want your identity to be public for this peer review? For information about this choice, including consent withdrawal, please see our Privacy Policy.

Reviewer #1: No

Reviewer #2: No

---

## [Decision Letter · Decision Letter 1]

16 Aug 2021

PONE-D-21-09645R1

Identifying Key Determinants and Dynamics of SARS-CoV-2/ACE2 Tight Interaction

PLOS ONE

Dear Dr. Jha,

Thank you for submitting your manuscript to PLOS ONE. After careful consideration, we feel that it has merit but does not fully meet PLOS ONE’s publication criteria as it currently stands. Therefore, we invite you to submit a revised version of the manuscript that addresses the points raised during the review process.

Reviewer 2 is supportive of accepting the manuscript in its revised form, but reviewer 1, as you will read,  has still many concerns. I am asking you to address these concerns in a revised version, and to write in a letter how you did  so, point  by point. I will then base my final decision on your edits and letter, I do not anticipate needing to send it again to the reviewers a thirds time. Please note that I am not necessarily requiring that you perform all requested additional calculations and statistical analysis. If you think these additional calculations are not needed, please indicate so and why in your letter. 

We look forward to receiving your revised manuscript.

Kind regards,

Jerome Baudry, Ph.D.

Academic Editor

PLOS ONE

Journal Requirements:

Additional Editor Comments (if provided):

Reviewers' comments:

Reviewer's Responses to Questions

**Comments to the Author**

1. If the authors have adequately addressed your comments raised in a previous round of review and you feel that this manuscript is now acceptable for publication, you may indicate that here to bypass the “Comments to the Author” section, enter your conflict of interest statement in the “Confidential to Editor” section, and submit your "Accept" recommendation.

Reviewer #1: (No Response)

Reviewer #2: All comments have been addressed

2. Is the manuscript technically sound, and do the data support the conclusions?

Reviewer #1: Partly

Reviewer #2: Partly

3. Has the statistical analysis been performed appropriately and rigorously? 

Reviewer #1: N/A

Reviewer #2: Yes

4. Have the authors made all data underlying the findings in their manuscript fully available?

Reviewer #1: Yes

Reviewer #2: Yes

5. Is the manuscript presented in an intelligible fashion and written in standard English?

Reviewer #1: Yes

Reviewer #2: Yes

6. Review Comments to the Author

Reviewer #1: The reviewer agrees that there are discrepancies in the glycosylation patterns, but not including any glycosylation in the structure does not address the issue. As it has been well established that glycosylation is a critical part of RBD-ACE2 complexes such that it affects the dynamics, more so in light of the paper: https://www.pnas.org/content/118/19/e2100425118, where they report that the glycans at positions N90 and N322 interact most strongly with the RBD and the N322 glycan of ACE2 interacts tightly with the spike RBD in 24 out of the 36 RBD–ACE2 complexes simulated, forming 5 to 10 interactions on average. With these published results, an un-glycosylated system provides little additional information. The authors mention that glycosylation occurs at a distance of 8-9Å and hence cannot influence that dynamics or binding, which is incorrect. There are long distance cross talks/interactions (i.e., Allostery) within proteins as well. On the other hand, the comparison between the FF is important and the study reports the importance of FF while running such simulations. Overall, the reviewer understands that this work was done earlier but with the findings available now, a biologically inaccurate model provides little insight into the SARS-Cov2 biology. However, the authors can consider focusing on FF comparison part of the paper rather than the biology. Alternatively, authors may compare their data with simulation from other groups (for example: https://doi.ccs.ornl.gov/ui/doi/98;
https://doi.ccs.ornl.gov/ui/doi/92) to make more coherent biological conclusions.

Reviewer #2: I consider the responses from the authors have meet the requirement of the Journal. All of my previous comments have been properly addressed.

7. PLOS authors have the option to publish the peer review history of their article (what does this mean?). If published, this will include your full peer review and any attached files.

Reviewer #1: No

Reviewer #2: No

---

## [Author Response · Author response to Decision Letter 1]

5 Sep 2021

We thank the reviewer for the valuable comment. Yes, our simulations did not include glycans as we depended only on the existing structural information for ACE2 during our work. We agree that in some computational studies (Mehdipour et al, PNAS, 2021, https://www.pnas.org/content/118/19/e2100425118), glycans have shown to be important for the interactions between RDB and ACE2. But we would also like to tell that other studies have failed to conclude any significant binding contribution from different forms of glycans on ACE2 with the spike protein (Nguyen et al, Viruses, 2021, https://doi.org/10.3390/v13050927 ). Quoting directly from Nguyen et al, “Notably, we find that this RBD-ACE2 contact signature is not altered by the presence of different glycoforms, suggesting that RBD-ACE2 interaction is robust”. 

While these studies took effort to go beyond the glycan information presented by different experimental structures (x-ray or cryo-EM), and performed simulations with a few mature, long and branched glycans (HT1 and HT2 by Mehdipour et al; MAN9 and FA2 glycans by Nguyen et al), the basis of selection of such glycosylation patterns for simulation is less intuitive or supported by experiments, and can become misleading on its own. A very recent work on comprehensive characterization of glycosylation on ACE2 (Shahjahan et al, Glycobiology, 2020, https://doi.org/10.1093/glycob/cwaa101 ) showed a large distribution of glycosylation patterns at ACE2 N322 position. The most populated glycan on N322 was bi-antennary (two branches) and smaller than the glycans used in simulation by Mehdipour et al and Nguyen et al in their studies (both used tri-antennary and/or chemically more complex glycan). Such differences in size and chemical complexity are further expected to reduce the impact on RBD-ACE2 interaction (Fig. 1).

Fig. 1. Discrepancy in choosing glycans on Asn322 (or N322) on ACE2. (A) Sahjahan et al showed a wide distribution of glycans on N322, with bi-antennary form to be most populated on ACE2. Curly brackets mean ambiguity of the subunit at any position in the glycan. (B) Mehdipour et al used various forms of glycan on N322 and observed tri-antennary glycoforms producing interactions with spike protein. (C) Nguyen et al used two forms of glycans on N322 of ACE2 in their MD simulations and observed no significant change in binding to spike protein.

A more recent publication by Allen et al (J. Mol. Biol, 2021, https://doi.org/10.1016/j.jmb.2020.166762 ) titled “Subtle influence of ACE2 glycan processing on SARS-COV-2 recognition” concluded from their experimental study that the ACE2 glycan may not influence SARS-CoV-2 binding. Directly quoting from the abstract, “We generated a panel of engineered ACE2 glycoforms which were analyzed by mass spectrometry to reveal the site-specific glycan modifications. We then probed the impact of ACE2 glycosylation on S binding and revealed a subtle sensitivity with hypersialylated or oligomannose-type glycans slightly impeding the inter- action. In contrast, deglycosylation of ACE2 did not influence SARS-CoV-2 binding. Overall, ACE2 glycosylation does not significantly influence viral spike binding. We suggest that any role of glycosylation in the pathobiology of SARS-CoV-2 will lie beyond its immediate impact of receptor glycosylation on virus binding.”

Hence, we expect our findings based on models that excluded glycans to be biophysically relevant.

We went ahead and cited the papers by Mehdipour et al, Nguyen et al and Allen et al in our manuscript, which will be useful for the readers to keep in mind while making conclusions from our study and applying to their own work. We also discuss this in the text (Conclusion section).

We checked out these data set https://doi.ccs.ornl.gov/ui/doi/92, which used CHARMM force field and https://doi.ccs.ornl.gov/ui/doi/98, which have many REMD simulations. We don’t know exactly what temperature ranges were used here. We believe that the REMD simulations of the RDB-ACE2 complex were also ran with CHARMM36 force field since there was the setup files with CHARMM36 for simulations run with Gromacs. If the simulations were run with GROMOS force field, we would love to examine them in our future studies.

We hope that our response to the comment is satisfactory and look forward to see our manuscript go to the next stage in the publication process. We are really thankful to both reviewers whose comments helped us probe further into the biophysics of spike protein – ACE2 interaction. Please, feel free to let us know any immediate deficiency in the manuscript which we can improve upon.

---

## [Editor Report · Decision Letter 2]

14 Sep 2021

Identifying Key Determinants and Dynamics of SARS-CoV-2/ACE2 Tight Interaction

PONE-D-21-09645R2

Dear Dr. Jha,

We’re pleased to inform you that your manuscript has been judged scientifically suitable for publication and will be formally accepted for publication once it meets all outstanding technical requirements.

Kind regards,

Jerome Baudry, Ph.D.

Academic Editor

PLOS ONE
---

## [Editor Report · Acceptance letter]

20 Sep 2021

PONE-D-21-09645R2 

Identifying key determinants and dynamics of SARS-cov-2/ACE2 tight interaction 

Dear Dr. Jha:

I'm pleased to inform you that your manuscript has been deemed suitable for publication in PLOS ONE. Congratulations! Your manuscript is now with our production department. 

Kind regards, 

on behalf of

Dr. Jerome Baudry 

Academic Editor

PLOS ONE